# A Method for Monthly Extreme Precipitation Forecasting with Physical Explanations

**Binlin Yang** [1,2] , **Lu Chen** [1,2,*] , **Vijay P. Singh** [3,4] , **Bin Yi** [1,2] , **Zhiyuan Leng** [1,2] , **Jie Zheng** [1,2] **and Qiao Song** [1,2]

1   School of Civil and Hydraulic Engineering, Huazhong University of Science and Technology, Wuhan 430074, China; bl_young@hust.edu.cn (B.Y.)
2   Hubei Key Laboratory of Digital Valley Science and Technology, Wuhan 430074, China
3   Department of Biological & Agricultural Engineering, and Zachry Department of Civil & Environmental Engineering, Texas A&M University, College Station, TX 77843-2117, USA
4   National Water and Energy Center, UAE University, Al Ain 31191-31195, United Arab Emirates
*   Correspondence: chen_lu@hust.edu.cn

**Abstract:** Monthly extreme precipitation (EP) forecasts are of vital importance in water resources management and storage behind dams. Machine learning (ML) is extensively used for forecasting monthly EP, and improvements in model performance have been a popular issue. The innovation of this study is summarized as follows. First, a distance correlation-Pearson correlation (DC-PC) method was proposed to identify the complex nonlinear relationship between global sea surface temperature (SST) and EP and select key input factors from SST. Second, a random forest (RF) model was used for forecasting monthly EP, and the physical mechanism of EP was obtained based on the feature importance (FI) of RF and DC–PC relationship. The middle and lower reaches of the Yangtze River (MLYR) were selected as a case study, and monthly EP in summer (June, July and August) was forecasted. Furthermore, the physical mechanism between key predictors with a large proportion of FI and EP was investigated. Results showed that the proposed model had high accuracy and robustness, in which $R^2$ in the test period was above 0.81, and *RMSE* as well as *MAE* were below 10 mm. Meanwhile, the key predictors in the high SST years could cause eastward extension of the South Asian High, westward extension of the Western Pacific Subtropical High, water vapor rising motion and an increase in the duration of atmospheric rivers exceeding 66 h, which lead to increasing EP in the MLYR. The results indicated that the DC–PC method could replace Pearson correlation for investigating the nonlinear relationship between SST and EP, as well as for selecting the factors. Further, the key predictors that account for a large proportion of FI can be used for explaining the physical mechanism of EP and directing forecasts.

**Keywords:** monthly extreme precipitation forecast; distance correlation; random forest; feature importance



## 1. Introduction

The frequency and intensity of extreme precipitation (EP) are reported to be increasing, as the global climate continues to warm and causes a variety of severe floods, flash flooding, urban waterlogging, and landslides [1,2]. These kinds of disasters often cause serious economic losses, ecological damage, and loss of life. For example, floods in 1998 caused USD 36 billion in economic losses and more than 3000 lives were lost in the Yangtze River valley in southern China and in the Nenjiang–Songhuajiang valley in northeast China [3]. EP in 2020 caused flash flooding and landslides, wreaking havoc across large areas of China, particularly along the Yangtze River [4]. Forecasting EP, therefore, is one of the most effective methods for the reduction of disaster losses, flood prevention, reduction of economic losses, and avoiding casualties [5]. Therefore, an investigation into EP prediction and its influencing factors is of great importance for the quantitative assessment of global or regional disaster and environmental risk [6–8].

RF is one of the most popular ML models for classification, prediction, studying variable importance, etc. [9]. The model has successfully emerged as an alternative forecasting method in some fields and obtained excellent results, such as daily and monthly rainfall prediction [10,11], flood prediction [12], and monthly EP indices prediction [13]. Herman et al. [14] explored the RF algorithm to forecast short-term EP in America, and found that the RF-based prediction was quite reliable. Wei et al. [15] performed seasonal predictions of EP based on RF and elucidated physical mechanisms of the EP event according to the decision trees in RF. In addition, the results of RF can be diagnosed using feature importance (FI). The FI of RF can provide the most useful predictive information and insights into the particular information for investigating the interpretability of a model [16]. For example, Łoś et al. [17] studied storm nowcasting with FI of the RF model and demonstrated that integrated water vapor (IWV) was the significant parameter for predicting storm location. Taken together with the previous study in RF, this indicated that the model is suitable for EP prediction, and FI can diagnose significant predictors and perform the physical interpretation of forecasting results. Therefore, RF was used to forecast EP in this study.

A problem for EP prediction is: What kinds of factors can be suitable predictors for the RF model? Previous studies forecasted rainfall using atmospheric circulation factors, monsoon system, plateau snow, Pacific subtropical high index, global sea surface temperature, etc. [18–22]. However, the main concern with the current models is that they are reliant on relatively robust relationships between the predictor and precipitation, which is not guaranteed in a changing weather system [23]. Lu et al. [24] demonstrated that summer precipitation had a robust nonlinear relationship with SST, which is predominantly quadratic. A multitude of studies have tried to identify skillful precipitation and EP predictors, and most of them ended up using the sea surface temperature (SST) [25].

The atmospheric circulation is driven by SST, which affects the distribution as well as intensity of precipitation and EP [26]. Furthermore, the SST is the source of moisture for precipitation and EP, and the variability of SST is an important signal that affects precipitation [27]. As a result, SST is often used for the development of precipitation and EP forecasts based on teleconnection methods [28], and Global SST holds potential for the prediction of precipitation and EP on an inter-annual time-scale [29]. For example, Chen and Georgakakos [28] obtained a new precipitation forecasting method by identifying SST "dipole" predictors, and the method was applied to the forecasting of seasonal precipitation over the southeast U.S.

It has been observed that RF does not perform well when it is applied to data sets with class noise [30], and the most important task for regional EP prediction is to choose the key input factors from the global SST. Fernando et al. [31] indicated that the task of an input selection algorithm is to determine the strength of the relationship between potential model input and output. There are many investigations on precipitation and EP prediction based on SST [32–36]. Many traditional approaches, such as Pearson correlation analysis, were used in the aforementioned studies to identify the potential linkages between SST and EP, and auto-regressive moving average and linear and nonlinear regression were used for precipitation or EP prediction. However, the relationship between regional EP and global SST is complex and nonlinear [37]. These methods are not robust enough to characterize the complex nonlinearity between EP and SST signals and cannot obtain preferable predictors [38].

To cope with the problem of nonlinearity and obtain the key input factors, there are two nonlinear measures of dependence, known as Kendall's tau and Spearman's rho. The disadvantage of the rank-based correlation coefficient is that there is loss of information when the data are converted to ranks; if the data are normally distributed, it is less powerful than the Pearson correlation coefficient [39]. Another is partial mutual information (PMI), which was proposed by Sharma [40]. The advantage of PMI is that it is model-free and uses a nonlinear measure of dependence (mutual information), which is often used to select inputs for ANN models [41]. The disadvantages of PMI are that (1) rainfall and SST are

continuous but the methods use the discrete version to calculate PMI, and (2) the method needs estimates of both marginal and joint probability distributions that are not suitable for the grid data of global SST [42].

The classical distance correlation proposed by Székely et al. [43] is a nonlinear measure of dependence between random vectors. The advantages of distance correlation are that it can illustrate the linear and nonlinear relationship of variables and does not have any model assumptions and parameter conditions [44]. The method has been applied to investigate the nonlinear relationship between air pollution and meteorological variables [45], gene–gene interactions [46], etc. Dalelane et al. [47] evaluated the global teleconnections in CMIP6 climate projections using the distance correlation. However, the method has not yet been used to describe the nonlinear relationship between SST and EP. Therefore, the distance correlation was applied to measure the relationship between SST and EP in this study. There is, however, a drawback with the distance correlation, which is that the value of the classical distance correlation is around 0 to 1, so this method cannot illustrate the positive and negative relationships between SST and EP for studying the physical mechanism of EP. To overcome this drawback, we first proposed the distance correlation-Pearson correlation analysis method (DC-PC), which can be used to explain the nonlinear relationship between global SST and EP for screening key input factors and explaining the physical mechanism of EP.

The objective of this paper was, therefore, to establish a new monthly EP forecasting model and investigate the physical mechanism of forecasting results. First, the DC-PC method was proposed to analyze the nonlinear relationship between global SST and EP. Second, we obtained the key input factors by the DC-PC method and forecasted EP based on the RF model. Third, the key predictors affecting the EP prediction were identified by the FI of the RF model. Finally, we explained the physical mechanism between key predictors and EP. The middle and lower reaches of the Yangtze River (MLYR) were selected as a case study. The innovation of this paper is given as follows. The DC-PC method was first used to identify the nonlinear relationship between global SST and EP, and the key input factors were obtained by the DC-PC method. Additionally, the key predictors affecting the EP prediction were identified based on the FI of RF. The main dynamical mechanism between the key predictors and EP was observed in MLYR, based on the DC-PC nonlinear relationship.

## 2. Materials and Methods

### 2.1. Input Selection of Random Forecast Model Based on DC-PC Method

The method was established on the basis of the distance correlation and the Pearson correlation coefficient. Different from the traditional calculation of the distance between sample moments, the sample distance correlation is to measure the degree of correlation between variables by calculating the Euclidean distance of the sample itself [43]. This study used the distance correlation coefficient to measure the relationship between EP and global SST.

Denoting two factors as $u$ and $v$, the distance correlation relationship is $\hat{d}corr(u,v)$. The sample $\{(u_i, v_i), i = 1, 2, \ldots, n\}$ is a random sample of the total sample $(u, v)$. Székely defined the distance correlation of $u$ and $v$ as

$$\hat{d}corr(u,v) = \frac{\hat{d}\mathrm{cov}(u,v)}{\sqrt{\hat{d}\mathrm{cov}(u,u)\hat{d}\mathrm{cov}(v,v)}} \tag{1}$$

where $\hat{d}\mathrm{cov}^2(u,v) = \hat{S}_1 + \hat{S}_2 - 2\hat{S}_3$.

$\hat{S}_1$, $\hat{S}_2$ and $\hat{S}_3$ are denoted as

$$\hat{S}_1 = \frac{1}{n^2} \sum_{i=1}^{n} \sum_{j=1}^{n} \|u_i - u_j\|_{d_u} \|v_i - v_j\|_{d_v} \tag{2}$$

$$\hat{S}_2 = \frac{1}{n^2}\sum_{i=1}^{n}\sum_{j=1}^{n}\|u_i - u_j\|_{d_u}\frac{1}{n^2}\sum_{i=1}^{n}\sum_{j=1}^{n}\|v_i - v_j\|_{d_v} \tag{3}$$

$$\hat{S}_3 = \frac{1}{n^2}\sum_{i=1}^{n}\sum_{j=1}^{n}\sum_{l=1}^{n}\|u_i - u_j\|_{d_u}\|v_i - v_j\|_{d_v} \tag{4}$$

where $i, j = 1, \ldots, n$; the same method can be used to calculate $\hat{d}corr(u, u)$ and $\hat{d}corr(v, v)$.

To solve the problem that distance correlation cannot show the positive and negative correlation, the Pearson correlation coefficient relationship was quoted to obtain the relationship between SST and EP, for which the data have passed the normal distribution test. The DC-PC correlation between SST and EP can be obtained based on Equation (5), which shows both the positive and negative relationship:

$$\hat{d}corr(u, v)_{pc} = \hat{d}corr(u, v)[r] \tag{5}$$

where $r$ is the Pearson correlation coefficient; and [] is the rounding symbol.

To select the key input factors and improve the accuracy of prediction, we established the test statistics $Z_n$ as shown in Equation (6) for testing the independence of random variables (Székely et al., 2007):

$$Z_n = \frac{n\hat{d}\text{cov}^2(u, v)}{\hat{S}_2} \tag{6}$$

We set the significance level $\alpha$ as 0.01, and the critical value as $\chi^2_{1-\alpha}$. The corresponding DC-PC value can be the input factor that passes the significance test ($p < 0.01$), while $Z_n \geq \chi^2_{1-\alpha}$.

### 2.2. Establishment of Prediction Model Based on RF

RF is a classification tree-based algorithm proposed by Breiman [48]. The algorithm diagram of RF regression is shown in Figure 1. The dataset of input factor $D$ is first randomly partitioned into $M$ groups, as $D_M$. Then, the predictions of $M$ single regression tree models are determined as $f(\mathbf{x})$. Further, the $M$ tree models are integrated to form the random forecast model $F(\mathbf{x})$ estimated by the average aggregation of base tree models [49]:

$$F(\mathbf{x}) = \frac{1}{M}\sum_{1}^{M}f_M(\mathbf{x}) \tag{7}$$

where $F(\mathbf{x})$ is the forecast result of RF; $\mathbf{x}$ is the input feature data vector; and $M$ is the number of regression tree models; $f(\mathbf{x})$ is the single regression tree model (Breiman 1984).

$$f(\mathbf{x}) = \sum_{t-1}^{t}C_l I(\mathbf{x} \in R_l) \tag{8}$$

where $R_l$ is the unit domain, which is segmented by the optimal segmentation variables, based on different features; $I(\mathbf{x} \in R_l)$ is the logic value, if $(\mathbf{x} \in R_l)$, $I(\mathbf{x} \in R_l) = 1$, else $I(\mathbf{x} \in R_l) = 0$; $C_l$ is the average of all output values contained in $R_l$; and $t$ is the cell field label.

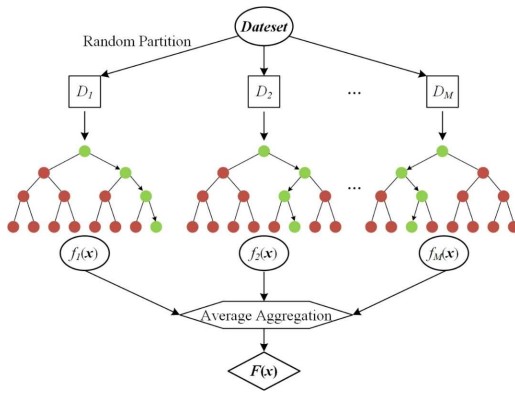

**Figure 1.** Algorithm diagram of RF regression.

### 2.3. FI Identification of the EP Prediction Model

RF has the advantages of high prediction accuracy, controllable generalization error, and fast convergence [50]; additionally, it gives the importance score of each feature. Feature extraction based on importance score has been widely used in medicine, economy, biology, and other fields [51–53]. The FI of the EP prediction model can be used to measure the impact of each input feature variable on EP. One of the methods used to measure the FI is the *Gini* index, which was used to measure the importance of predictors in this paper.

First, the purity of the model at the split node $k$ can be calculated by

$$Gini(p_k) = 1 - \sum_{k=1}^{N_K} p_k^2 \tag{9}$$

where $N_K$ is the number of categories, and $p_k$ is the weight of the $k$ categories.

The feature $f_i$ is used as the classification basis of $k$ and can be measured according to the *Gini* index of branches. The purity of feature $f_i$ at the split node $k$ can be calculated by

$$Gini_{f_i,k} = Gini(p_k) - Gini(p_l) - Gini(p_r) \tag{10}$$

where $Gini(p_l)$ and $Gini(p_r)$ represent the *Gini* index of left and right branches after branching, respectively, and the importance of features is:

$$VIM_{f_i} = \sum_{j=1}^{m} VIM_j^{Gini} \tag{11}$$

$$VIM_j^{Gini} = \sum_{m \in M} Gini_{f_i,k} \tag{12}$$

where $VIM_{f_i}$ is the importance of $f_i$ at the $M$ trees; $VIM_j^{Gini}$ represents the importance of $f_i$ in the decision tree of $j$; and $m$ represents the single tree.

### 2.4. Performance Evaluation of the Proposed Model

The explained variance scores ($EVS$), $R^2$, mean absolute error ($MAE$), root mean square error ($RMSE$), and forecast pass rate ($P_r$) can be used to comprehensively evaluate the performance of the proposed model.

$EVS$ indicates the similarity between the predicted value and the historical value, as shown in Equation (13). If $y_i = \hat{y}_i$, the explained variance is 1. Otherwise, the smaller the explained variance is, the less accurate the prediction is.

$$EVS(y_i, \hat{y}_i) = 1 - \frac{Var\{y_i - \hat{y}_i\}}{Var\{y_i\}} \tag{13}$$

where $y_i$ is the corresponding target output; $\hat{y}_i$ is the estimated target output; and *Var* is the square of the standard deviation.

$R^2$ is the coefficient of determination that represents the percentage of variance in the historical values that can be explained by simulations.

*MAE* represents the absolute prediction error sampled across all samples. The unit of mean absolute error is consistent with the unit of the dependent variable. The closer it is to 0, the more accurate the model is, which is given as follows.

*RMSE* is the standard deviation of residuals between predictions and observations. The value of *RMSE* ranges between 0 and ∞; the smaller the *RMSE*, the better the accuracy.

$$R^2 = 1 - \frac{\sum\limits_{i=1}^{N} (y_i - \hat{y}_i)^2}{\sum\limits_{i=1}^{N} (y_i - \overline{y})^2} \tag{14}$$

$$\overline{y} = \frac{1}{N}\sum_{i=1}^{N} y_i \tag{15}$$

$$MAE = \frac{\sum\limits_{i=1}^{N} |\hat{y}_i - y_i|}{N} \tag{16}$$

$$RMSE = \sqrt{\frac{\sum\limits_{i=1}^{N} (\hat{y}_i - y_i)^2}{N}} \tag{17}$$

where $N$ is the sample length; $\hat{y}_i$ and $y_i$ are the predicated and historical values for the year $i$, respectively.

The predicate rate $(P_r)$ was used to evaluate the accuracy of the prediction model. Results of prediction are eligible, while the error of prediction is less than 20%.

$$P_r = \frac{A}{N} \times 100\% \tag{18}$$

where $A$ is the eligible sample length.

## 3. Data

The daily precipitation data in summer (June, July, and August) of the MLYR from 1979 to 2020 were obtained from the Meteorological Data Center of the China Meteorological Administration. The locations of study catchments and 99 meteorological stations in the MLYR are shown in Figure 2. Accounting for regional differences, the 95th percentile was used to define the EP threshold [54]. The previous study found that SST in winter (December of previous year, January, and February) and spring (March, April, and May) has a significant relationship with the EP of MLYR in summer [55]. Therefore, the monthly extended reconstructed SST version 4 data set from 1978 to 2010 with a resolution of 2.5° × 2.5° was collected from the National Oceanic and Atmospheric Administration (NOAA) [56]. The EP series from 1979 to 2010 was selected for the training set. The series of EP from 2011 to 2020 was the test set for model prediction. In addition, in the mechanism analysis section, monthly 200- and 500-hpa geopotential height as well as 1000 to 300-hpa omega field from 1979 to 2010 with a resolution of 2.5° × 2.5° were obtained from the National Centers for Environmental Prediction (NCEP)/National Center for Atmospheric Research (NCEP/NCAR) reanalysis data set [57]. To identify the ARs of MLYR, 6-hourly specific humidity and $u$ and $v$ components of wind components with high resolution (0.25° × 0.25°) from the ERA-5 reanalysis project were used. All of the cited ERA-5 data cover a 20°–40° N, 90°–140° E area from 1979 to 2010 at 20 vertical pressure levels.

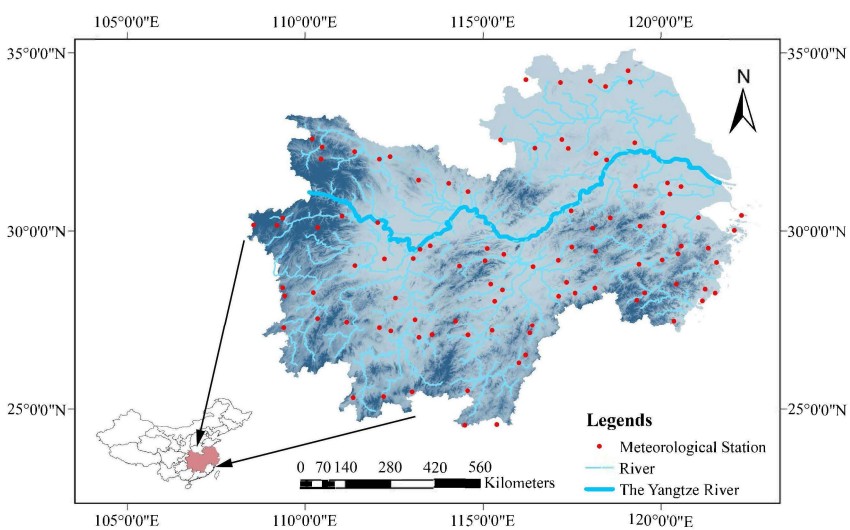

**Figure 2.** Location of the catchments and 99 meteorological stations.

## 4. Prediction of EP

### 4.1. Identification of Input Factors

The nonlinear relationship between global SST and EP in MLYR were calculated from 1978 to 2010 using the DC-PC method. Results are shown in Figure 3. It can be seen from Figure 3 that the more inside the area is, the stronger the DC-PC relationship between EP and SST is. The areas labeled with white dots represent the significantly correlation relationship between EP and SST. Here, we chose the significant area as the key input factors of the RF model.

As shown in Figure 3, in June, EP had a positive correlation with the northern Atlantic Ocean SST in year-ago December (NAO-Dec), the southern China Sea in January (SCS-Jan), the northern Indian Ocean in January (NIO-Jan), the southern Atlantic Ocean SST in April (SAO-Apr), and the southern Indian Ocean SST in April as well as May (SIO-Apr, SIO-May). The EP in June was negatively correlated with the southern Atlantic Ocean SST in year-ago December (SAO-Dec) and the Atlantic Ocean in March (AO-Mar).

The EP occurring in July had a positive correlation with the eastern Pacific Ocean in year-ago December, February, March, May, and June (EPO-Dec, EPO-Feb, EPO-Mar, EPO-May, EPO-Jun). Meanwhile, the positive correlation between EP in July and the northern Indian Ocean SST was increasing as the month increased from March to June (NIO-Mar, NIO-Apr, NIO-May, NIO-Jun). In addition, the northwestern Pacific Ocean SST (NWP-Apr) had a positive correlation with EP in July.

The EP in August had a positive correlation with the mid-southern Pacific SST in year-ago December, February, March, and April (MSP-Dec, MSP-Feb, MSP-Mar, MSP-Apr). The northeastern Pacific SST in January and February (NEP-Jan, NEP-Feb) was positively correlated with EP occurring in August. Moreover, the EP in August showed a positive correlation with the northern Atlantic Ocean in year-ago December (NAO-Dec), the southeastern China Sea SST in June (SECS-Jun), and the northwestern Pacific SST in May as well as June (NWP-May, NWP-Jun). By contrast, the EP in August had a negative correlation with the southeastern Pacific Ocean SST in year-ago December, January, February, and March (SEP-Dec, SEP-Jan, SEP-Feb, SEP-Mar). The mid-eastern Pacific Ocean SSTs in year-ago December and February (MEP-Dec, MEP-Feb) were negatively correlated with the EP in August. Meanwhile, the southern Atlantic Ocean in January and February (SAO-Jan, SAO-Feb) had a negative correlation with the EP occurring in August.

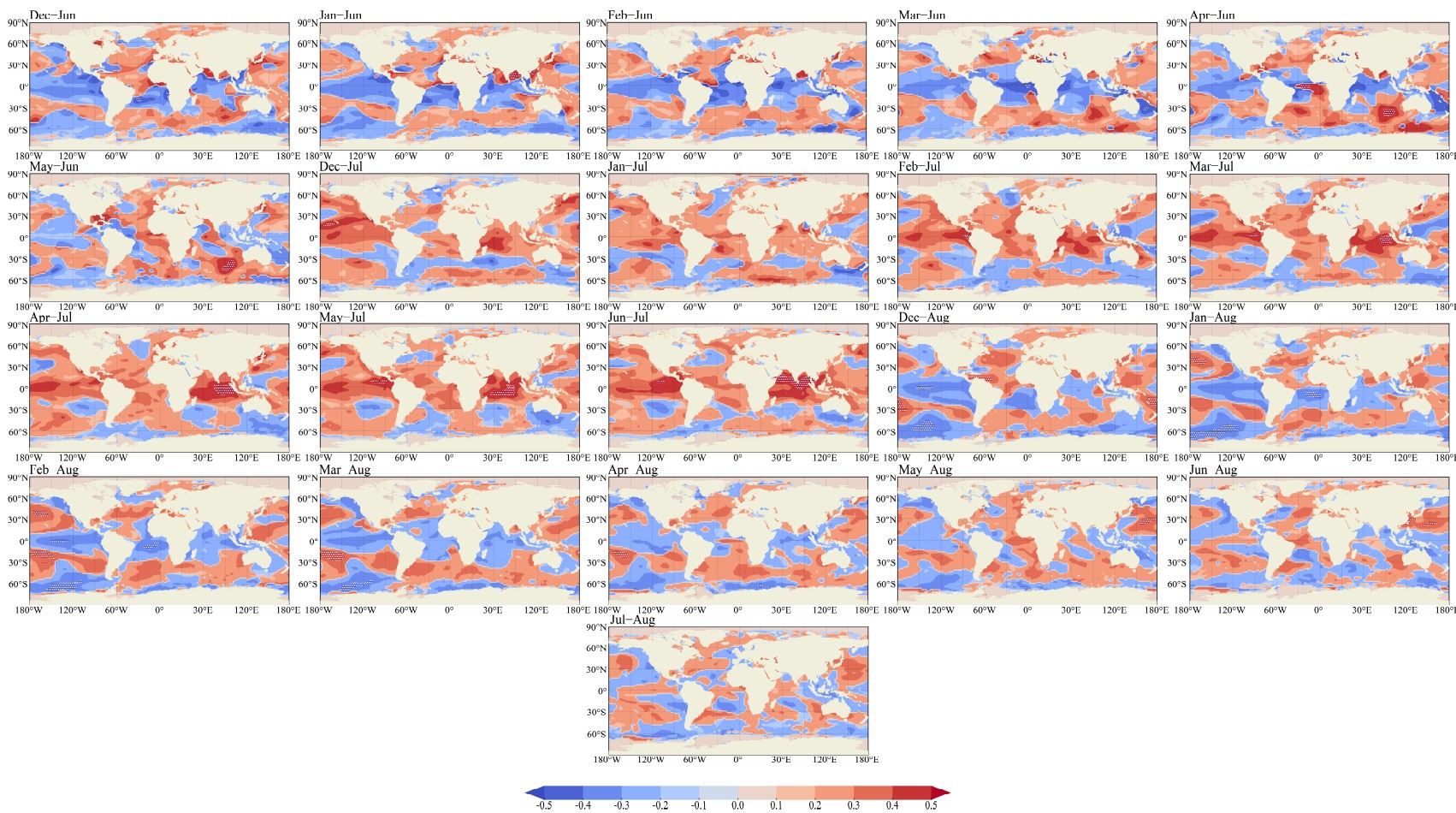

**Figure 3.** DC-PC relationship between global SST and summer EP in the MLYR. (The pictures from left to right represent the relationship between the EP in June and SSTs from December to May, the EP in July and SSTs from December to June, as well as the EP in August and SSTs from December to July, respectively; the areas labeled with white dots in the pictures represent the significantly correlation).

According to the DC-PC relationship between global SST and the summer EP in MLYR, the significant areas were selected as the key input factors of the RF model, as shown in Table 1.

**Table 1.** Key input factors of RF model.

| Month | Key Input Factors of Model |
|---|---|
| June | NAO-Dec, SAO-Dec, SCS-Jan, NIO-Jan, AO-Mar, SAO-Apr, SIO-Apr, SIO-May |
| July | EPO-Dec, EPO-Feb, EPO-Mar, NIO-Mar, NIO-Apr, NWP-Apr, NIO-May, EPO-May, NIO-Jun, EPO-Jun |
| August | MSP-Dec, MEP-Dec, NAO-Dec, SEP-Dec, SAO-Jan, SEP-Jan, NEP-Jan, MSP-Feb, NEP-Feb, MEP-Feb, SAO-Feb, SEP-Feb, MSP-Mar, SEP-Mar, MSP-Apr, NWP-May, SECS-Jun, NWP-Jun |

Previous studies have shown that the winter Atlantic Ocean SST affected rainfall in MLYR by stimulating the Eurasian Rossby Wave Trains, propagating from the northern Atlantic Ocean to the east of the Urals [58]. The warm Indian Ocean SST can lead to anormal cyclone response over the Philippine Sea and MLYR, leading to anormal precipitation in MLYR [59]. These results indicated that the Atlantic Ocean and Indian Ocean SSTs had a strong relationship with EP in MLYR. Zhao et al. [60] found that the early winter SST in the Middle and East Pacific had an important influence on the July precipitation in the Jianghuai region of China, which is consistent with the results for the DC-PC correlation in July.

*4.2. Forecast Results of the Proposed Model*

The EP forecast results in June, July, and August are shown in Table 2 and Figure 4. As shown in Table 2, the performance in the training period was satisfactory, where $R^2$ was above 0.97, and *RMSE* as well as *MAE* were below 3 mm. $R^2$, *EVS* and $P_r$ in the test period were above 0.81, and the *RMSE* and *MAE* were below 10 mm. These results indicate that the proposed method had high accuracy and strong robustness. The prediction result of the proposed model had an improvement about 0.3 over that of the previous study [35], where the study area and the resolution of data were basically the same. Wu et al. [61] predicted monthly rainfall in the upper and middle Yangtze River basin using the multipole SST anomaly model (MSST), and the August prediction result was lower than that for June and July. Compared to results of Wu et al. [61], the prediction results of this study were steadier than those from the MSST-PFMS model.

**Table 2.** Evaluation of the proposed model.

| Month | Training Period | | | | | Test Period | | | | |
|---|---|---|---|---|---|---|---|---|---|---|
| | $R^2$ | EVS | RMSE | MAE | $P_r$ | $R^2$ | EVS | RMSE | MAE | $P_r$ |
| June | 0.99 | 0.99 | 1.94 | 1.33 | 100% | 0.87 | 0.91 | 7.58 | 5.96 | 90% |
| July | 0.99 | 0.99 | 2.34 | 1.27 | 100% | 0.81 | 0.85 | 9.37 | 7.02 | 80% |
| August | 0.97 | 0.97 | 2.77 | 1.90 | 100% | 0.83 | 0.87 | 8.35 | 6.28 | 90% |

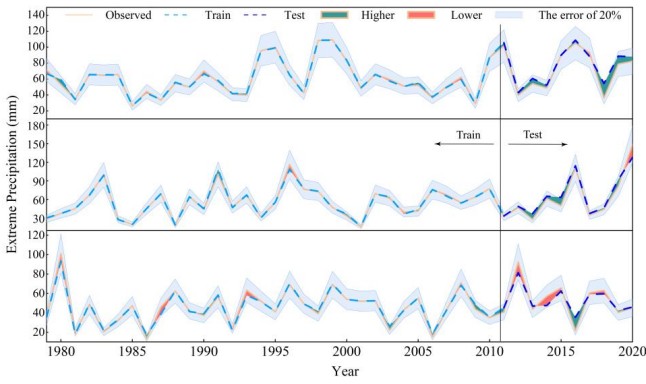

**Figure 4.** Prediction performance results of the proposed method during training and testing.

### 4.3. FI of Predictors

The FI of RF can provide superior means for measuring the feature relevance of data, which can increase the interpretability of a model [62], and the FI of the key input factors is shown in Figure 5. In June, the FI proportion of NAO-Dec was 51%, accounting for more than half of all factors. The previous study also showed that the relationship between the summer rainfall anomalies in MLYR and SST anomalies in the Atlantic Ocean had a significant correlation [63].

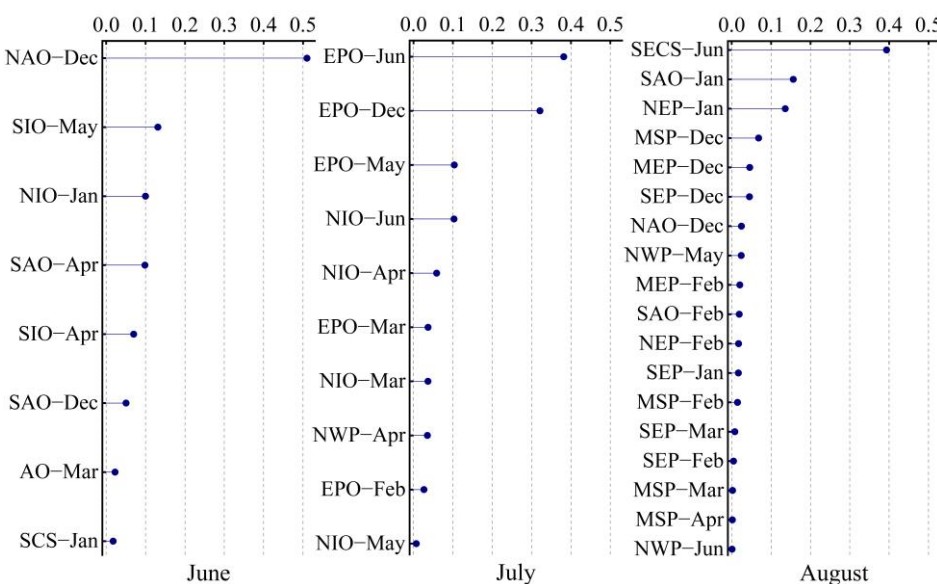

**Figure 5.** FI of RF in June, July, and August.

In July, the FI proportions of EPO-Dec and EPO-Jun were 0.39 and 0.32, respectively. The locations of EPO-Dec and EPO-Jun were in the area of ENSO 3. Li et al. [64] indicated that increases in the ENSO 3 index generally led to a significant increase in EP in the MLYR in the following year, and the effect was reflected in December of the current year and January of the following year. Rong et al. [65] also found that ENSO events could increase rainfall in MLYR when the ENSO 3 index increased in the current year.

In August, the FI proportion of SECS-Jun in the forecast model was 0.39. The location of SECS was in the northern Kuroshio area. Yu [66] indicated that there was a strong teleconnection correlation between the northern Kuroshio area in the East China Sea and the rainfall in MLYR. Li et al. [67] found that there was a significant positive correlation between SST in the northern Kuroshio area from April to June and the rainfall in MLYR, and the temperature in the Kuroshio area could be a predictor of MLYR precipitation.

In summary, NAO-Dec, EPO-Dec, EPO-Jun, and SECS-Jun, which have a positive correlation with EP, can be key predictors of the model for EP prediction in MLYR.

## 5. Physical Mechanism of EP in MLYR

### 5.1. Discussion of EP Occurring in SST Anomaly Years

To obtain and confirm the physical connection between predictors of the model and EP in MLYR, we investigated the key predictors (NAO-Dec, EPO-Dec, EPO-Jun, and SECS-Jun), which accounted for a large proportion of the model and had a positive correlation with EP. The regional key predictor SSTs with area weighting were calculated to obtain the standardized monthly time series during 1978–2010. The high SST years were defined as months with key predictor SSTs greater than one standard deviation, and low SST years were defined as months with key predictor SSTs lower than one standard deviation [68]. Results are shown in Figure 6. In June, four NAO-Dec SST high years and seven NAO-Dec low SST years were selected for the study. In July, three EPO-Dec high SST years and four

EPO-Jun high SST years were greater than one standard deviation and could be selected. Additionally, four EPO-Dec low SST years and three EPO-Jun low SST years were chosen for the study, with lower than one standard deviation. In August, six SECS-Jun high SST years and five SECS-Jun low SST years were screened. Further, the differences in the geopotential height, water vapor vertical motion, and the intensity as well as duration of ARs during 1979–2010 in MLYR were compared with high and low SST years.

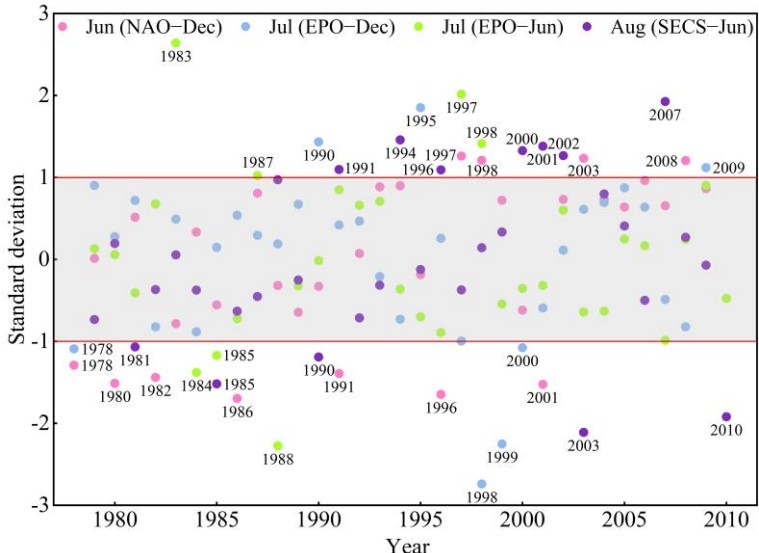

**Figure 6.** Years with high and low SST anomalies.

To ensure that the physical mechanisms addressed in the study were valid, the monthly average EP during summers occurring in high SST years was compared with that occurring in low SST years from 1979 to 2010, as shown in Figure 7. The anomalous NAO-Dec SST affected EP the following year in June. The EP occurring in July was affected by the anomalous EPO-Dec SST and EPO-Jun SST. In addition, the anomalous SECS-Jun SST affected the EP in August. As can be seen from Figure 7, the EP in high SST years exceeded that in low SST years. The differences between EP occurring in high and low SST years in July were the largest. In NAO-Dec, EPO-Dec, EPO-Jun and SECS-Jun high anomaly years, the average EP was 19.06 mm, 66.82 mm, 57.63 mm, and 21.82 mm more than that in the low anomaly years, respectively. These results indicated that summer EP increased in the high SST years.

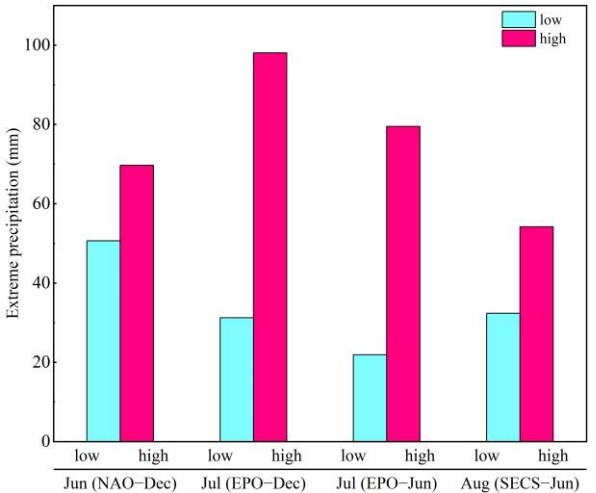

**Figure 7.** Monthly mean EP in SST anormal years of the MLYR from 1979 to 2010.

### 5.2. Comparison of Geopotential Height in SST Anomaly Years

To investigate the atmospheric circulation characteristics of EP in MLYR, the 200- and 500-hpa geopotential height during the years with high SST and low SST were compared, as shown in Figures 8 and 9. As shown in Figure 8, the location of the South Asian High (SAH) was analyzed with the 12,600 gpm of the geopotential height at the 200-hpa level. In NAO-Dec, EPO-Dec, as well as EPO-Jun high SST years, it was found that the SAH extended to the east, compared with that in the low SST years, during June and July. In August, the SAH moved over MLYR in the SECS-Jun high SST years, and moved on to the North China Plain in the SECS-Jun low SST years. The geopotential height analyses at the 500-hpa level are shown in Figure 9. In June and July, the Western Pacific Subtropical High (WPSH) defined as 5880 gpm extended to the west in the NAO-Dec, EPO-Dec, as well as EPO-Jun high SST years compared to the low SST years. In August, WPSH moved over the MLYR and northern China in the SECS-Jun high and low SSTs years, respectively.

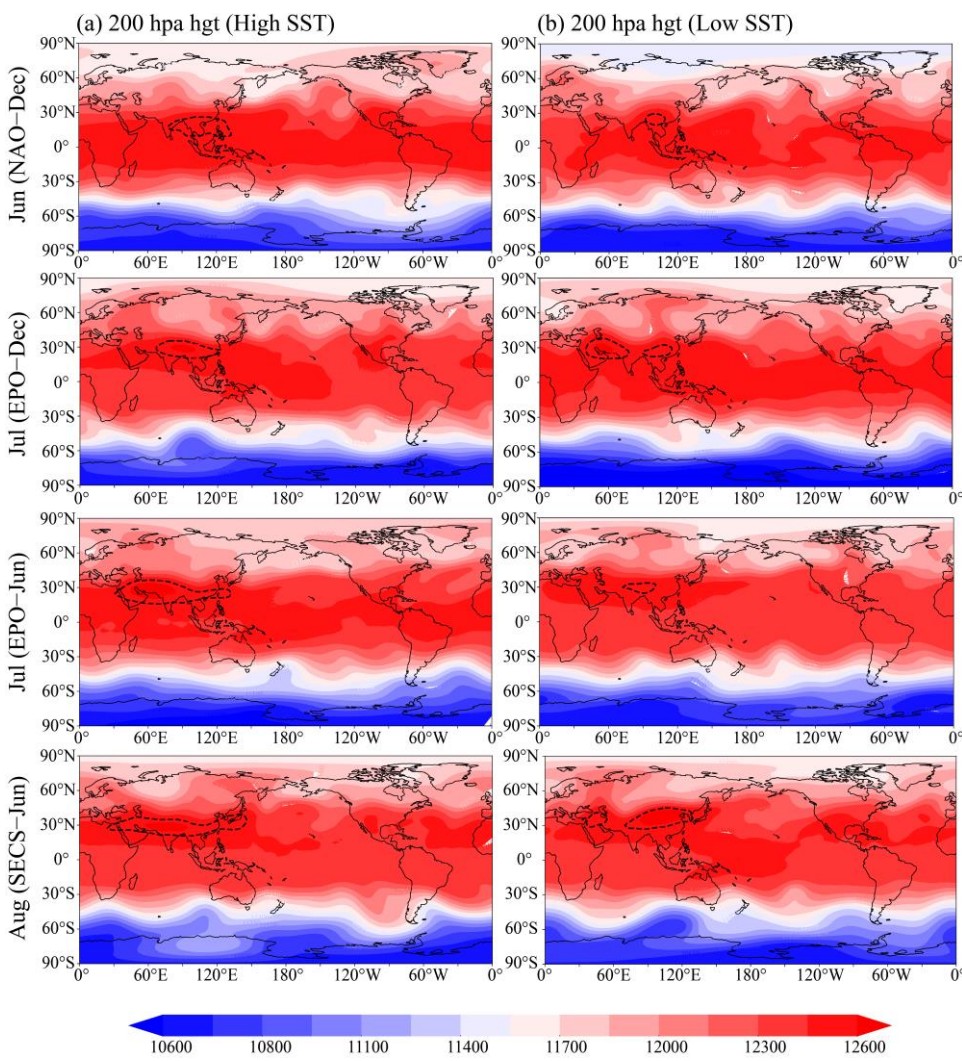

**Figure 8.** The 200-hpa geopotential height of the summer MLYR in SST anormal years. The black dotted lines define the 12,600 gpm; (**a**,**b**) mark the geopotential height (unit: gpm) anomalies with high SSTs and low SSTs, respectively.

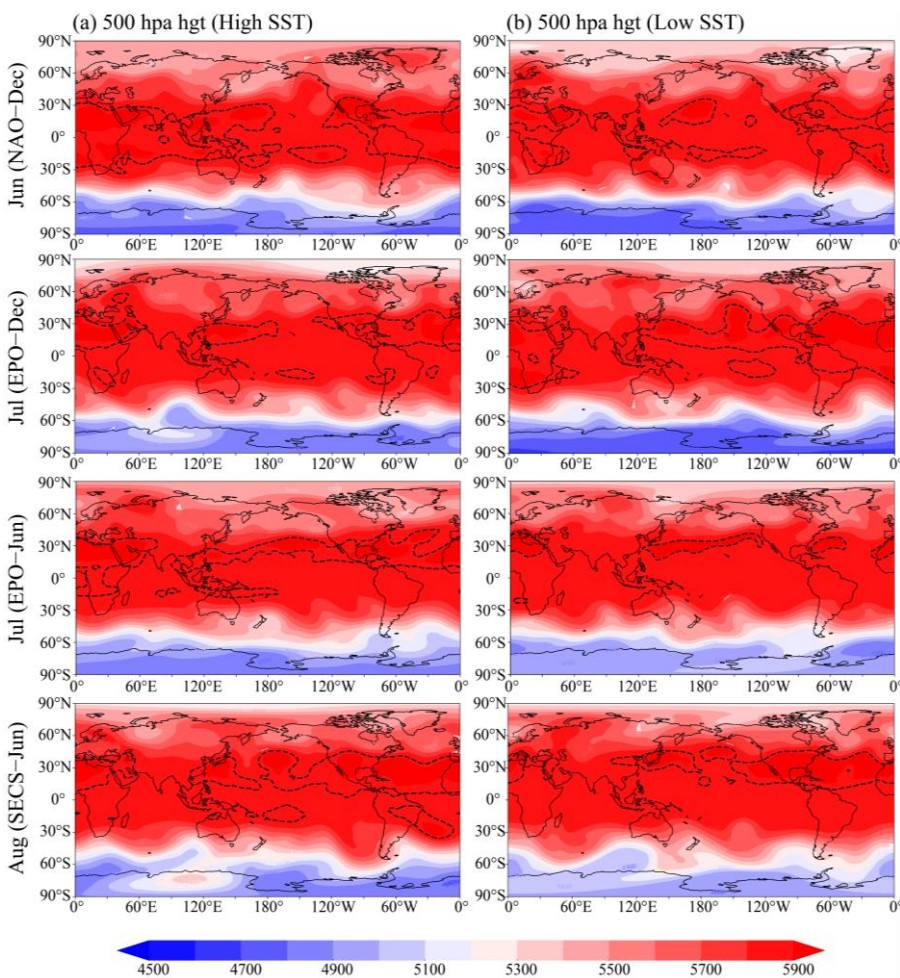

**Figure 9.** The 500-hpa geopotential height of the summer MLYR in SST anormal years. The black dotted lines define the 5880 gpm; (**a**,**b**) mark the geopotential height (unit: gpm) anomalies with high SSTs and low SSTs, respectively.

Chen et al. [69] noted that the SAH anomaly extended to the east on the 10–30 d sub-seasonal scale, was often accompanied by the abnormal westward extension of the 588 line of the subtropical high, and led to abnormal precipitation in the MLYR. At the same time, the westward extension of WPSH could cause the confluence of cold and warm currents on the northern side of the western Pacific, which was beneficial for the formation of heavy rainfall [70]. In this study, the positions of the WPSH were located in the southern MLYR during June to July, which could lead to the northern cold front and the southern warm front meeting, with increasing precipitation. Hong et al. [71] showed that the warming SST over the northern Atlantic Ocean (NAO) could continue from the winter of the previous year to the summer of the next year, and the positive (negative) SST anomalies in the tropical NAO could lead to stronger (weaker) WPSH. There was a significant positive correlation between the area and the intensity of the WPSH and SST in the eastern Pacific Ocean (EPO) [72]. Furthermore, when the SST of the Kuroshio area located north of the monthly average subtropical high ridge was high, the WPSH moved over the MLYR [73]. The previous studies indicated that the key predictor SST could lead to the eastward extension of the SAH and westward extension of the WPSH, and then lead to increased EP.

*5.3. Comparison of Water Vapor Vertical Motion in SST Anomaly Years*

We calculated the water vapor vertical velocity from 1000 to 300 hpa during 1979 to 2010 in the MLYR. The effects of key predictors on water vapor vertical motion were analyzed by subtracting the vertical velocity fields of low SST years from the vertical velocity fields of high SST years. Results are shown in Figure 10, where the blue marks the water vapor rising motion, and orange marks the water vapor sinking motion. In June, with the effect of NAO-Dec SST, the water vapor vertical motion was a negative anomaly in 111°E–123°E of the MLYR. In July, the negative anomaly covered all of the MLYR with the effect of EPO-Dec SST. Furthermore, the negative anomaly centered north of the MLYR, affected by the EPO-Jun SST. In August, the negative anomaly covered the whole MLYR for the disturbance of SECS-Jun SST. These observations collectively demonstrate that the high SST years of NAO-Dec, EPO-Dec, EPO-Jun, and SECS-Jun can increase the eastward extension of the SAH and the westward extension of the WPSH, which can reinforce the water vapor coagulation and lead to increased EP in the MLYR.

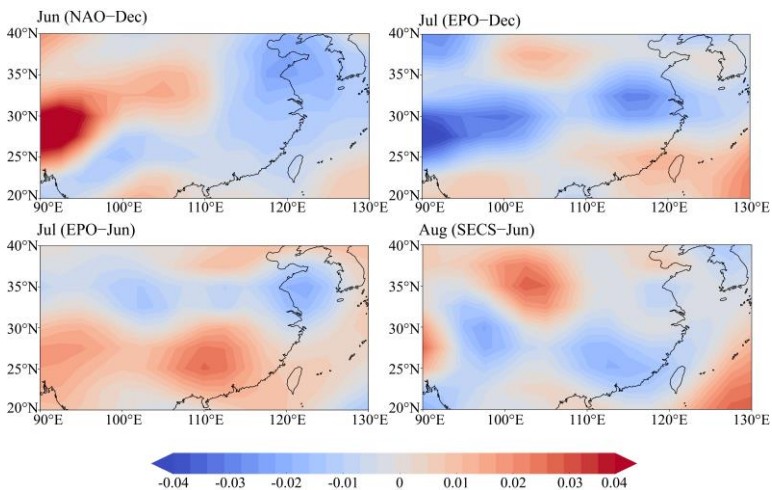

**Figure 10.** Difference in water vapor vertical velocity fields (unit: Pa/s) between summers with the high and low SST years.

*5.4. Comparison of ARs in SST Anomaly Years*

To investigate the characteristics of water vapor transport during the SST anormal years in the summer MLYR, we calculated two indexes (duration and intensity) of ARs that passed though the MLYR in summer from the year 1979 to the year 2010. First, the atmospheric river events occurring in the MLYR were identified according to the tracking algorithm [74]. Second, the cumulative weighted average duration and intensity of ARs that passed through the MLYR are compared in the high and low SSTs years in Figure 11. As shown in Figure 11, in the NAO-Dec, EPO-Dec, EPO-Jun and SECS-Jun high SST years, all durations of ARs in summer were 587.1 h, 354 h, 258 h and 98.5 h longer than in the low SST years, respectively. Especially, in the high SST years, the durations of ARs exceeding 66 h were longer than the low SST years. In the summer MLYR, the AR intensity in the high SST years was generally higher than in the low SST years, and the AR intensity in SECS-Jun high and low years was inverse, where the difference was not significant.

Ding et al. [75] confirmed that EP in many regions in China, such as the Yangtze River basin and Yellow River basin, was related to the ARs. Xiong et al. [76] found that the EP in China caused by ARs accounted for 70–90% of the total precipitation and up to 90% of the EP over the MLYR. As discussed by Ralph et al. [77] and Lamjiri et al. [78], long-duration atmospheric river events can contribute to the accumulation of large precipitation totals. The study in this paper also found that the durations of ARs were the main factor increasing EP in the MLYR, and the long-duration ARs were more conducive to EP.

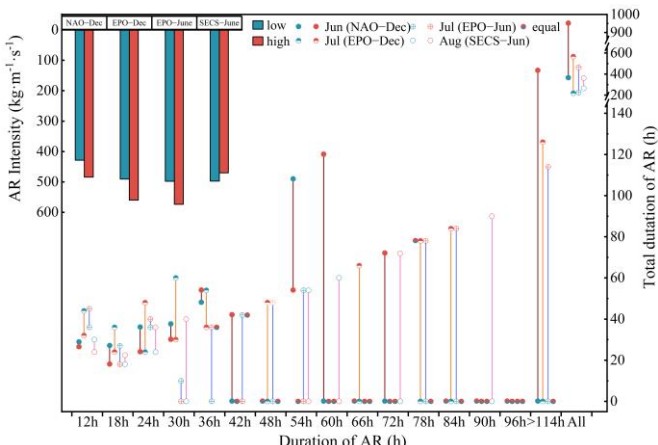

**Figure 11.** Total duration (unit: h) and intensity (unit: kg·m$^{-1}$·s$^{-1}$) of ARs that passed through the MLYR in the summer in SST anormal years.

## 6. Conclusions

A long-term extreme rainfall prediction method, considering model interpretability and the nonlinear relationship between inputs and outputs, was proposed. The proposed method allowed comprehensive observation of the complex nonlinear relationship between global SST and EP, which can be used for forecasting as well as selecting key predictors to explain the physical mechanism of EP. The middle and lower reaches of the Yangtze River were selected as a case study. The key input factors of the summer EP in the MLYR were obtained by the DC-PC method. The RF model was used to forecast the monthly EP in the MLYR. The obtained key predictors also could be used to explain the physical mechanism of EP. The main conclusions are summarized as follows.

(1) The DC-PC model can identify the complex nonlinear relationship between SST and EP and provide the key input factors for the RF model. The prediction results of the RF model indicate that the key input factors proved by the proposed method can be used for forecasting monthly EP preferably, and the accuracy as well as robustness of the model are better than those in previous studies.

(2) The proposed method can be applied to the prediction of summer EP in the MLYR, and the key predictors can be used to explain the physical mechanism of EP. $R^2$ in the calibration period was above 0.97, and *RMSE* as well as *MAE* were below 3 mm. In the test period, $R^2$ was above 0.81, and the *RMSE* and *MAE* were below 10 mm. The FI of RF indicated that NAO-Dec, EPO-Dec, EPO-Jun, and SECS-Jun are good indicators for the prediction of summer EP in the MLYR. Discussion of the physical mechanism demonstrated that the high SST of key predictors can cause an eastward extension of the SAH, westward extension of the WPSH, water vapor rising motion and long-duration ARs, which lead to increasing EP in the MLYR.

(3) This method also can be extended to forecast regional rainfall using SST data in the future. Additionally, the method can identify key predictors of regional precipitation and explain the physical mechanism of EP based on these predictors. However, the meteorological and hydrological time series are generally considered nonstationary changes as the global temperature increases. With the nonstationary changes of the series, the relationship between SST and EP will become more complex, and the DC-PC method can be improved in order to analyze the complex relationships and select key predictors for prediction tasks.

**Author Contributions:** L.C.: Conceptualization, Writing—Review and Editing. B.Y. (Binlin Yang): Methodology, Writing—Original Draft, Software, Formal analysis. V.P.S.: Writing—Review and Editing. B.Y. (Bin Yi): Software, Investigation. Z.L.: Investigation, Formal analysis. J.Z.: Software. Q.S.: Data Curation. All authors have read and agreed to the published version of the manuscript.

**Funding:** This research is supported by the National Key Research and Development Program of China (2021YFC3200400), the Science and Technology Plan Projects of Tibet Autonomous Region (XZ202301YD0044C).

**Data Availability Statement:** The precipitation data can be obtained from the Meteorological Data Center of the China Meteorological Administration (http://data.cma.cn/, accessed on 1 July 2021). The monthly extended reconstructed SST version 4 data set can be collected from the National Oceanic and Atmospheric Administration (NOAA, https://www.psl.noaa.gov/, accessed on 15 January 2022). The monthly 200- and 500-hpa geopotential height as well as 1000 to 300-hpa omega field can be obtained from the National Centers for Environmental Prediction (NCEP)/National Center for Atmospheric Research (NCEP/NCAR) reanalysis data set (https://www.weather.gov/ncep/, accessed on 10 May 2022).

**Acknowledgments:** The authors are thankful for the support from Texas A&M University.

**Conflicts of Interest:** The authors declare no conflict of interest.

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
