# Peer review of "A Method for Monthly Extreme Precipitation Forecasting with Physical Explanations"

_water, doi:10.3390/w15081545_

Round 1
Reviewer 1 Report
no comment
Author Response
Dear Reviewer: Thank you very much for taking the time to review the manuscript. We carefully refined the manuscript during the first round of review. We hope you find our revised manuscript to your satisfaction. We will be happy to address any further concerns or questions you may have. Thank you again for your time and effort in reviewing our manuscript.

Reviewer 2 Report
This article report your interesting work to estimate the monthly extreme precipitation using maching learning method and taking data of Sea Surface Temperature as main predictor.
In some figures of the paper I found hard to read the label of the axis and legends since the letters are small compare with text letters. If you agree you can improve it.
Author Response
Dear Reviewer:
Thank you very much for your time involved in reviewing the manuscript. We would like to express our sincerest gratitude for your valuable comments and suggestions on our manuscript. Our response is listed in the attachment.

Reviewer 3 Report
The article examines and evaluates a new method in predicting precipitation and finally river flow, and in my opinion, it has good innovation and valuable results that can be used by researchers. The article is acceptable in its current state.
1. What is the main question addressed by the research? To test a new method for xtreme precipitation forecast. 2. Do you consider the topic original or relevant in the field? Does it address a specific gap in the field? We have various methods for forecaste of extreme precipitation and introduces of new methods also necessary and this article present a new method. 3. What does it add to the subject area compared with other published material? Introduce and evaluation of a new method. 4. What specific improvements should the authors consider regarding the methodology? What further controls should be considered? To test of efficiencyand accuracy of new method all framework needed considered. 5. Are the conclusions consistent with the evidence and arguments presented and do they address the main question posed? 6. Are the references appropriate? Yes appropriate. 7. Please include any additional comments on the tables and figures. No correction needed.
Author Response
Dear Reviewer:
Thank you very much for your time involved in reviewing the manuscript. We have carefully improved the manuscript in round 1 of review. We hope that you find our revised manuscript satisfactory. And we would be happy to address any further concerns or questions that you may have. Thank you once again for your time and efforts in reviewing our manuscript.

Reviewer 4 Report
The content of this article is suitable for the scope of the Water journal. However, before a final decision there some points that need better clarification. It is hoped that the authors can respond to the following comments with plausible responses.
Comments
1) The title can be changed to “A Method for Monthly Extreme Precipitation Forecast with Physical Explanations” There is no appreciable novelty in the text, but mixture of well-known methodologies,
2) Introduction section is very lengthy. Can be reduced almost to half.
3) Abstract: Extreme precipitation is important also for water management and storage behind dams.
4) Page 3, line 128: What is meant by "Pearson correlation analysis"? This correlation method is used in different stochastic processes like Markov and ARIMA processes. The authors must specify the role of this correlation analysis. Besides, Pearson correlation is valid for normally (Gaussian) probability distribution function (PDF). Did the authors checked the normality (Gaussian) of the data or not. This point must be elaborated in the article text.
5) Page 3, line 145: What is the difference between distance correlation and cross-correlation. Can the authors use cross-correlation for distances?
6) Page 5, line 191: As said before the authors should check the validity of Pearson correlation coefficient, because it needs Gaussian PDF of the data?
7) Page 6, Eq.14İ Is not this expression square of the Pearson correlation definition?
8) Page 7: Section "2.5 Physical Mechanism of Key Predictors" does not seen fitting all what have been explained above. It can be deleted without any harm to the whole text.
9) Page 9, Table 2: R2 values are equal to 1 practically, which is very surprising, but in the abstract its value is mentioned as 0.81, so where from does the difference comes. This is given in Table 3. The calibration period is only for ten years, and therefore reduction in R2 values in the projection part of the data could be effected by small samples compared to calibration period.
10) Page 14, lines 396-398: What is the significance of one standard deviation. It is useful in case of normal (Gaussian) PDF of the data.
11) Page 18, line 522: The words "nonuniformity change" is not traditionally known neither in meteorology nor in hydrology literature. Perhaps they mean trend component. In this case are the trends of increasing or decreasing type.
12) The last sentence in the Conclusions section is an allegation without evidence.
Author Response
Dear Reviewer: Thank you very much for taking the time to review the manuscript. We would like to express our sincere thanks for your valuable comments and suggestions on our manuscript. Your insightful feedback and constructive criticism have greatly helped us improve the quality of our research work. Our responses are set out in the annex.
